



# META3.1exp : A new Global Mesoscale Eddy Trajectories Atlas derived from altimetry

Cori Pegliasco[1], Antoine Delepoulle[1], Rosemary Morrow[2], Yannice Faugère[1], Gerald Dibarboure[3]

[1]CLS, Ramonville Saint Agne, 31250, France
[2]LEGOS, Toulouse, 31400, France
[3]CNES, 18 Avenue Edouard Belin, Toulouse, 31400, France

*Correspondence to*: Cori Pegliasco (cpegliasco@groupcls.com)

**Abstract.** This paper presents the new global Mesoscale Eddy Trajectories Atlases (META3.1exp DT all-satellites, https://doi.org/10.24400/527896/a01-2021.001, Pegliasco et al., 2021a and META3.1exp DT two-satellites, https://doi.org/10.24400/527896/a01-2021.002, Pegliasco et al., 2021b), composed of the eddies'

identifications and trajectories produced with altimetric maps. The detection method used is a heritage of the py-eddy-tracker algorithm developed by Mason et al. (2014), optimized to manage with efficiency large datasets, and thus long time series. These products are an improvement of the META2.0 product, produced by SSALTO/DUACS and distributed by AVISO+ (https://aviso.altimetry.fr) with support from CNES, in collaboration with Oregon State University with support from NASA and based on Chelton et al. (2011).

META3.1exp provides supplementary information such as the mesoscale eddy shapes with the eddy edges and their maximum speed contour, and the eddy speed profiles from the center to the edge. The tracking algorithm used is based on overlapping contours, includes virtual observations and acts as a filter with respect to the shortest trajectories. The absolute dynamic topography field is now used for eddy detection, instead of the sea level anomaly maps, to better represent the ocean dynamics in the more energetic areas and close to coasts and

islands.

To evaluate the impact of the changes from META2.0 to META3.1exp, a comparison methodology has been applied. The similarity coefficient is based on the ratio between the eddies' overlap and their cumulative area, and allows an extensive comparison of the different datasets in terms of geographic distribution, statistics over the main physical characteristics, changes in the lifetime of the trajectories, etc. After evaluating the impact of

each change separately, we conclude that the major differences between META3.1exp and META2.0 are due to the change in the detection algorithm. META3.1exp contains smaller eddies and trajectories lasting at least 10 days that were not available in the distributed META2.0 product. Nevertheless, 55% of the structures in META2.0 are similar in META3.1exp, ensuring the continuity between the two products, and the physical characteristics of the common eddies are close. Geographically, the eddy distribution mainly differs in the

strong current regions, where the mean dynamic topography gradients are sharp. The additional information on the eddy contours allows more accurate collocation of mesoscale structures with data from other sources, so META3.1exp is recommended for multi-disciplinary applications.





## 1 Introduction

Mesoscale eddies are ubiquitous in the global ocean. Ranging from tens to hundreds of kilometers and spanning days to years (Morrow and Le Traon, 2012), mesoscale eddies strongly participate in the redistribution of energy, heat and salt in the ocean, and other bio and chemical components (Beal et al., 2011; Chaigneau et al., 2011; Gaube et al., 2014; Gruber et al., 2011; Zhang et al., 2014). The physical characteristics of eddies are also coupled with biologic data to infer the behaviour of marine animals (Braun et al., 2019; Chambault et al., 2019;

Christie et al., 2010; Siegel et al., 2008; Staaterman et al., 2012), and can be used to track pollutants (Brach et al., 2018; Gilchrist et al., 2020). Over the past three decades, the development of altimetry maps with an increasing accuracy has made their observation at global scale possible.

Nowadays, lots of methods have been developed to detect and track eddies. The difficulty in defining a mesoscale eddy is linked to its separation from the background oceanic field. Eddies are mainly generated by

current instabilities, or from ocean instabilities due to wind or topographic obstacles, creating variability around the ocean's mean state. As such, they are often considered as anomalies. Moreover, in a homogeneous background, rotating structures are associated with high and low pressure or sea level surfaces through the geostrophic equilibrium. Thus, a large number of studies detect eddies in Sea Level Anomaly (SLA) maps where anticyclones (respectively, cyclones) are associated to areas with positive (negative, respectively)

anomalies, delimited in space by geometric criteria (Chaigneau et al., 2008, 2009; Faghmous et al., 2015; Liu et al., 2016). Other studies are more interested in the rotation of the coherent structures to separate them from a non-rotative background with the Okubo-Weiss parameter, or by rotational speed consideration (Isern-Fontanet et al., 2003; Le Vu et al., 2018; Mkhinini et al., 2014; Morrow et al., 2004; Nencioli et al., 2010).

Many regional studies on the detection and tracking of mesoscale eddies have been conducted, but global

analyses are rare due to the lack of globally accessible databases of mesoscale eddies. It is simpler to properly tune a detection algorithm in a restricted area, where such characteristics as the Earth's rotation or the ocean stratification are homogeneous, than to take into account their variability at a global scale (Zhang et al., 2013). It is also faster, less consuming of computing capacities and easier to manipulate reasonable quantities of data, concentrated in time and space. As mesoscale eddies generated by the destabilization of strong currents are

different from island generated eddies for example, and because the water masses differ from an oceanic basin to another, a regional approach was often chosen to better integrate the eddies' specificities.

Nevertheless, some global databases of mesoscale eddies exist (Chelton et al., 2011, 2007; Faghmous et al., 2015; Martínez-Moreno et al., 2019; Tian et al., 2020; Zhang et al., 2013). The first global database was presented in Chelton et al. (2011), covering the 1993 – 2008 period (hereafter CH11). This database was

regularly updated until 2016 to consider the extending period, the changes of the input altimetry maps (weekly then daily production, improvement of the standards) and was available on the Oregon State University (OSU) affiliated webpage (http://wombat.coas.oregonstate.edu/eddies/index.html). The production of the database was then transferred to CLS/CNES team in 2017 and since then is updated operationally and distributed by AVISO as the Mesoscale Eddy Trajectories Atlas (META1.0exp in 2017) thanks to a fruitful collaboration between

OSU and CLS/CNES. In 2018, META2.0 was produced with an improved tracking scheme. The introduction of virtual observations manages the "missing eddy problem", replacing eddies absent in consecutive maps if altimetric tracks do not cross the structure for several days. A better management of coasts and islands, in



particular in the numerous archipelagos, was also developed (META2.0 handbook, SALP-MU-P-EA-23126-CLS, 2020). Both META1.0exp and META2.0 detections were made on spatially filtered SLA maps, with

geometrical consideration to determine the eddies' interior. The access to the eddy contours was indirect, approximated by a circle defined by the centroid and the speed radius of the eddy. The delivered trajectories had a minimum lifetime of 28 days.

We aim here to present a new version of the daily global Mesoscale Eddy Trajectories Atlas (META3.1exp), distributed by AVISO. The main changes with this new system are that the eddy detection is made on filtered

ADT maps with geometrical detection method, the main atlas provides eddy trajectories lasting more than 10 days, but also identifies the trajectories shorter than 10 days, and the lone eddies which are identified only one day. The source code is written in Python and is made freely available under a GPL V3 license (https://github.com/AntSimi/py-eddy-tracker), inherited from the Py-Eddy-Tracker (PET) of Mason et al. (2014) and improved after a fruitful collaboration between E. Mason (IMEDEA) and CLS. This new atlas is available

in two versions using the ADT maps detections: with two-satellites, the T/P-Jason and ERS-Envisat-Saral dual sampling giving the most consistent spatial coverage over the full time series, and all-satellites using all available missions for the best possible sampling coverage, albeit that varies over time.

This paper compares this new product to the current AVISO operational version META2.0. Such a comparison is complex due to the size of the datasets and the diversity of variables describing the mesoscale eddies within

similarly organized atlases. The originality of the methodology we developed is its capacity to match eddies from different atlases as well as comparing the global distributions of parameters. The Similarity Coefficient (SC) employed here is based on the ratio between the common area of two eddies and their total area. The algorithm is able to rapidly process the large global atlases over the whole altimetry period. After pairing the eddies between the two datasets, we separate 1) the common and similar eddies which have the highest SCs, 2)

the common but different eddies with moderate SCs, 3) the eddies only present in one dataset and thus presenting a novelty with regards to the other dataset, and 4) the eddies with multiple matches. These groups of eddies provide not only statistics about the detected eddies, but also insights about how the associated trajectories are distributed.

Four major changes are made in the evolution of the processing from META2.0 to META3.1exp. First, the

change in the detection algorithm from the historical OSU code (Chelton et al., 2011) to PET algorithm (Mason et al., 2014) impacts significantly on the number of detected eddies and their surface characteristics (amplitude, radius). Secondly, the tracking scheme was modified to use the eddies' overlap in successive daily maps instead of searching for new eddy candidates in a restricted area, increasing the length of many trajectories. Thirdly, we have changed the input fields from SLA to ADT maps, in order to better identify eddies in energetic regions

with strong Sea Surface Height (SSH) gradients and where recurrent mesoscale structures exist, either as eddies or meanders. Many geographically correlated eddies are formed near coasts, bathymetric changes, from wind-orographic effects, or current retroflections, and have an imprint in the Mean Dynamic Topography (MDT). Using the ADT rather than SLA field provides a more consistent eddy detection and presence in such oceanic regions. Finally, the preprocessing step of filtering, inherent of the OSU and PET methodologies, was modified

to better separate the eddies from the background large-scale ocean circulation. We produced intermediate datasets to independently assess how each change impacts on the atlas of eddy trajectories.



This paper is organized as follows. In Sect. 2, we present first the altimetry gridded maps used to detect eddies, the differences and similarities between the OSU and PET detection and tracking algorithms, and the methodology developed to efficiently compare the different atlases. In Sect. 3, we illustrate how each change made from META2.0 to META3.1exp impacts on the detected eddies and their statistics, and present some META3.1exp characteristics with regards to META2.0. We document the availability of META3.1exp in Sect. 4 and provide a summary in Sect. 5.

## 2 Data and methods

### 2.1 Altimetric fields for eddy detection

In the continuity of the CH11 dataset, the META2.0 product is based on the SLA maps produced by the Copernicus Climate Change Service (C3S) and distributed in the C3S Climate Data Store (https://cds.climate.copernicus.eu/). These maps are built using at most two altimetric missions, with the Topex-Poseidon and Jason satellites on the same long-term ground tracks, and a second satellite mission, mainly on the ERS-Envisat-Saral-Sentinel-3A ground tracks. As the sampling and the represented scales are stable throughout time, this dataset is considered to be homogeneous in time in terms of climate signals and mesoscale content.

The META3.1exp has been computed as well from the C3S dataset to ensure continuity for the users but a second production, based on the Copernicus Marine Environment Monitoring Service (CMEMS, marine.copernicus.eu) products built from the complete altimetric constellation has been performed. The all-satellites merged product is built with all the available satellites at a given time, improving the small scales representation in the maps due to the diversity of the tracks' location and the different repetition periods of the altimetric missions (Pascual et al., 2006). The mesoscale eddies retrieved from these products are thus improved at small scales.

Beside the input constellation, the products have an additional difference regarding the temporal mean reference used to compute anomalies. For the two-satellites product, the SLA is obtained as the difference between the along track instantaneous Sea Surface Height (SSH) and the Mean Sea Surface (MSS), the gridded proxy derived from all the available altimetry missions. For the all-satellite product, SLA is obtained as the difference between the SSH and the Mean Profiles (MP), the most precise MSS, available on the long-term repeat tracks (for details see Pujol et al., 2016). This different strategy to remove a MSS will also have slight impact on the eddy field.

The META3.1exp product is based on the ADT. The ADT is the sum of the SLA and the Mean Dynamic Topography (MDT), the later corresponding to the mean oceanic circulation derived from multiple satellite and in-situ data. The accuracy of the MDT has greatly improved in recent years, giving robustness to the ADT fields (Rio et al., 2014).

The input fields (SLA or ADT, C3S or CMEMS) are all global daily products, with a ¼° grid resolution, using the most recent reprocessing version (DT2018, Taburet et al., 2019). The along-track data are filtered with a low pass Lanczos filter depending on the latitude (250 km near the Equator, down to 55 km at high latitudes) and subsampled at 14 km (Pujol et al., 2016). Each daily map is produced with an optimal interpolation using spatial and temporal decorrelations scales varying with latitude (Pujol et al., 2016). When several satellites are used, a weight is attributed to each mission to take into account their noise and thus, their confidence level.



The mapping procedure tends to filter out the smaller scales, but the effective resolution of the DT2018 global maps was estimated to range from ∼100 km wavelength at high latitude to ∼800 km wavelength in the equatorial band, meaning that ∼25 km radius structures are properly resolved at high latitudes, ∼200 km radius structures are resolved in the equatorial band and ∼50 km radius structures are resolved at mid-latitudes (Ballarotta et al., 2019).

**2.2 Detection and tracking algorithms : OSU and PET**

**2.2.1 Detecting mesoscale eddies**

The META2.0 detection algorithm (OSU) is similar to Chelton et al. (2011), based on the fact that closed contours of SLA correspond approximately to the streamlines of a geostrophic flow. The method aims to find a geographic region of connected pixels having all SLA values below (or above) the local maximum (or

minimum) SLA value for anticyclonic (or cyclonic) eddies. Several SLA extrema are authorized within one eddy region. There is a 1 cm threshold for amplitude (the absolute SSH difference between the edge and the extremum of the structure), a maximum of 1000 pixels within a structure, and gaps between pixels in longitude and latitude are not allowed. These restrictions avoid the detection of ameba-like regions as eddies, because eddies are expected to show more compact form to maintain their rotation. The eddy center location provided in

META2.0 is the centroid of the SSH of the connected pixels. The eddy radius is the radius of the circle that has the same area as the region within the closed contour of SSH with maximum averaged speed. The effective radius, computed for tracking purposes but not delivered, is the radius of the circle with the same area as the connected pixels. Eddies can thus be represented by the location of their center and by their speed radius. No detection is made within ± 2.5° latitude of the equator, due to the non-geostrophic balance near the equator.

The OSU detection differs from the original Chelton et al. (2011) methodology as the minimum of 8 consecutive pixels is not required, since we found that the smallest amplitude criteria prevents the detection of structures with unrealistic small radii. Note that the SLA fields are filtered before the eddy-detection in order to remove the large scale anomaly patterns, such as the chevron like patterns near the equator or the El Niño induced displacement of warm water from west to east in the Equatorial Pacific, following the procedures of

Chelton et al. (2011). The filtering step is made in OSU with a 2D Lanczos filter, with a 1000 km half-power cutoff wavelength in latitude and longitude to take into account the latitudinal variation of the dimension of a grid pixel.

For the META3.1exp dataset, the detection algorithm is based on the Mason et al. (2014) py-eddy-tracker (PET) algorithm, loosely based on Chelton et al. (2011), Kurian et al. (2011) and Penven et al. (2005). Working either

on SLA or ADT fields, the SSH contours are interpolated instead of using the SSH pixels. The SSH fields are also high pass filtered for META3.1exp, although we changed the half-power cutoff wavelength of the 2D Lanczos filter, setting it to 700 km. More details on this choice are provided in Sect. 3.1.1. Eddy detection is made by scanning closed contours from SSH maxima downward for Anticyclonic Eddies (AEs) and from SSH minima upward for Cyclonic Eddies (CEs). For the outermost closed contour encompassing only one extremum,

a shape error test is performed. This test, similar to Kurian et al. (2011) verifies that the ratio between the areal sum deviations of the contour from its best fit circle and the area of this best fit circle is below a certain value. This specification aims to avoid the selection of eddies with shapes too different from circles, where rotation is



not possible, as for banana shapes for example. In Mason et al. (2014) and Kurian et al. (2011), the shape error was limited to 55%. We increase this value to 70% to ensure that elongated eddies are detected, a case often

visible in highly dynamic regions and when eddies are interacting.

The META3.1exp dataset includes both the effective contour (outermost closed contour) and the speed contour (contour with the maximum averaged speed around it). The effective radius is deduced from the best-fit circle applied to the effective contour; similarly, the speed radius is derived from the best-fit circle applied to the speed contour. We chose to decrease the amplitude threshold from 1 cm to 0.4 cm, since with a minimum of 5

consecutive pixels within a contour we ensure a minimum geographic imprint instead of limiting the amplitude parameter. As only one extremum is accepted within an eddy, contrary to OSU, the position of this extremum is provided, but the location of the center is deduced from the best-fit circle of the speed contour. The change in the number of extrema allowed within an eddy contour impacts strongly on the eddy detection and the eddy's characteristics, and will be presented with more details in Sect. 3.1.3.

We also provide a new characteristic, the mean speed profile between the effective contour and SSH extremum, as proposed by the AMEDA detection and tracking algorithm from Le Vu et al. (2018). This speed profile is useful for dynamical investigations and comparisons with theoretical eddy shapes. Specifying the speed and effective contours are very helpful for the colocation of altimetric-derived eddies with external data, such as Sea Surface Temperature (SST), ocean color, surface salinity, winds, and in situ measures as Argo floats, XBT or

CTDs, Niskin bottle sampling, larvae presence, since they allow a more precise anisotropic positioning of the mesoscale eddies.

To ease the manipulation of these large files, the speed profiles and the contour data are regularly interpolated over 50 evenly spaced points.

### 2.2.2 Tracking

The META2.0 dataset is composed of trajectories lasting more than 28 days, following the four weeks minimum eddy duration of Chelton et al. (2011). This limitation was linked to the weekly availability of the maps, and is close to the mean temporal resolution of 34 days of the altimetry maps (Ballarotta et al., 2019). The tracking procedure consists of searching for an eddy at the time step $t + dt$ (dt = 1 day) in a restricted area around the center of the eddy considered at $t$. After analyzing the mean displacement of eddies in the previous

META1.0exp, the restricted area was set to evolve with latitude. From high latitudes towards 25° in latitude, the radius of the restricted area is set to 50 km, then a progressive increase is made to reach a radius of 100 km at 10° latitude, since the eddies are larger and travel faster in the equatorial band. Searching within the restricted area prevents the association of unrelated eddies resulting in large jumps within a trajectory. The variation of the candidate eddy size at $t + dt$ (amplitude and effective radius) must fall between 0.4 and 2.5 times the reference

eddy size at $dt$. If several candidates are found at $t + dt$, the eddy added to the trajectory is the one minimizing a cost function based on the distance between the centers.

As for other tracking procedures (Chaigneau et al., 2008; Faghmous et al., 2015; Laxenaire et al., 2018; Le Vu et al., 2018; Li et al., 2016; Pegliasco et al., 2020), we have to deal with the "missing eddy" problem, i.e., the disappearance of an eddy for some days between altimetric groundtracks in the mapped SSH fields, or due to

restrictions imposed by the detection procedure. This "missing eddy problem" is solved by authorizing the research of a new candidate eddy over several days, we chose four days. To consider the eddy's displacement,



the radius of the search area is increased by ⅓ each supplementary day (Figure 1a). If after 1, 2, 3 or 4 days a candidate is available, it is associated to the trajectory. The days without eddies are flagged to identify a virtual eddy, whose characteristics are interpolated from the two detected eddies. Thus, there are at most three

consecutive virtual observations over the 4-day gaps. Due to the extension of the search area, candidates may be found that cross land. To avoid this, a land management checks if the core of the eddies, represented as the fifth of their respective speed radius, is able to move from one eddy to another without crossing land (Figure 1b). When crossing land, the eddy tracking association is not allowed. If after 4 days of research, no candidates are found, the trajectory is stopped.


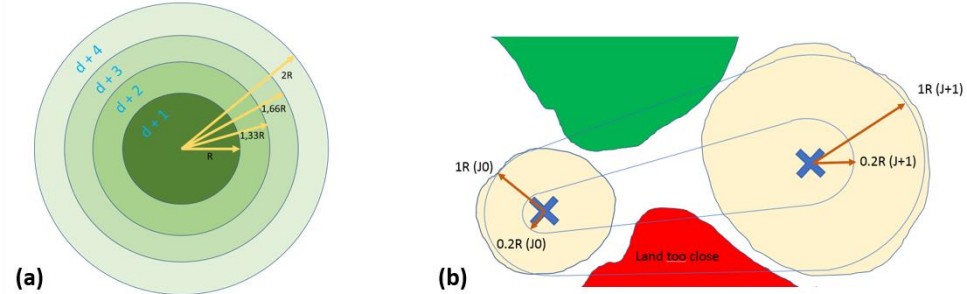

**Figure 1: Tracking parameters in META2.0. (a) Increasing the restricted area to solve the missing eddy problem. (b) Land management to avoid eddies crossing land.**

The tracking procedure used in META3.1exp is different, based on the overlap of the effective contours and not

on the search over a restricted area. Tracking by overlap has shown its robustness with daily mapped data (Keppler et al., 2018; Laxenaire et al., 2018; Li et al., 2016; Pegliasco et al., 2020, 2015), since the day-to-day general displacement of mesoscale eddies does not exceed 10 km. Here eddy candidate is retained if the overlap ratio, defined as the ratio between the overlapping area and the union of the two eddies' area, is more than 5%. This allows us to track a small eddy included in a large eddy (or the reverse situation) due to eddy splitting (or

merging). Note that 95% of the eddy associations have an overlap ratio greater than 20%. No restrictions are imposed on the radius or amplitude variations, since we observed that in the case of merging of two small eddies into one, or splitting of one large eddy in two, the rapid variation of the radius ends prematurely one trajectory and start a new one in META2.0. In the case of several candidate eddies, the larger overlap ratio is retained. Even if the merging and splitting events are not recorded in the tracking procedure for META3.1exp, the

overlap method ensures the continuity of the trajectory when merging or splitting events occurs. The tracking procedure allows us to search for a candidate eddy for up to 5 days instead of 4 (thus, a maximum of 4 consecutive virtual observations), since the overlap ensures a geographic proximity. This proximity was the only criterion in the cost function used in META2.0 in case of several candidates, whose number increases significantly when the radius of the search area and the time increase.

To test the ability of this new tracking to produce robust trajectories, we investigate the characteristics of the trajectories obtained with the two different methods (restricted area and overlap) from a similar detection, made with the PET algorithm on the two-satellites ADT maps high-pass filtered with a 700 km wavelength cutoff. Table 1 shows that more observations are associated with trajectories lasting at least 10 days when the overlap



method is employed than with the restricted area method (~91% vs ~87%). The observations within trajectories
shorter than 10 days and the untracked eddies are less numerous with the overlap method. The number of virtual
observations introduced consecutively is four for the overlap tracking and three for the restricted area tracking;
nevertheless, fewer virtual eddies are needed in trajectories with the overlap tracking than with the restricted
area tracking.

| Tracking method | Eddies in trajectories lasting at least 10 days | Eddies in trajectories lasting less than 10 days | Eddies detected but Untracked | Virtual eddies in trajectories lasting at least 10 days | Virtual eddies in trajectories lasting less than 10 days |
|---|---|---|---|---|---|
| Overlap | 91.1% | 7.7% | 1.2% | 5.4% | 4.9% |
| Restricted area | 87.0% | 10.3% | 2.7% | 5.8% | 6.3% |

**Table 1: Repartition of eddies after tracking made by the overlap or the restricted area methods. Note that percentages are obtained relatively to the sum of the observations within trajectories ≥ 10days, < 10 days and the untracked eddies for the first three columns, and relatively to the number of observations in the trajectories ≥ 10 days or < 10 days, respectively, for the last columns.**

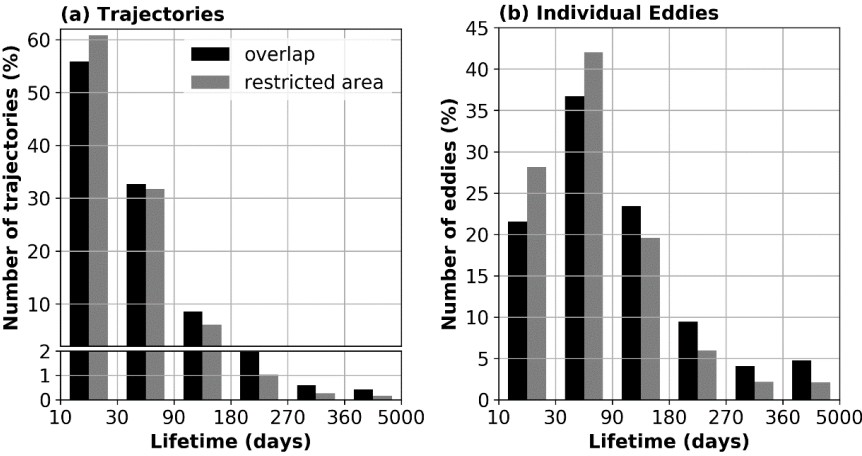

**Figure 2 : Distribution of the number of (a) trajectories and (b) individual eddies by lifetime for the restricted area (grey) and the overlap (black) tracking methods built from the same detection dataset. Percentages are relative to the number of trajectories lasting at least 10 days and the individual eddies for each tracking : 1.67 million trajectories built with the restricted area tracking from 62 million individual eddies and 1.46 million trajectories built with the overlap tracking from 65 million individual eddies. Note the discontinuity in the yaxis of (a).**

When evaluating the distribution of the number of trajectories for different lifetimes (Figure 2a), we should keep
in mind that the majority of the trajectories built have lifetimes between 10 – 30 days, whether we use the
overlap method (55 %) or the restricted area method (60 %) but only involved a small number of individual
eddies (21 % for the overlap; 28 % for the restricted area) (Figure 2b). This implies that the analyses made on
the remaining trajectories is representative of only half of the trajectories but concerns 70 – 80 % of the
individual eddies. Whatever the tracking method, the trajectories lasting more than 6 months represent only a
few percent of the dataset, but the overlap method detects twice as many very long trajectories than the
restricted area method. Note that the occurrence of four consecutive virtual eddies in the case of the overlap



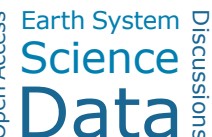

tracking is stable over different eddy lifetimes, so longer trajectories do not particularly rely on the presence of more virtual eddies.

We can thus conclude that the overlap tracking is able to efficiently associate detected eddies into long trajectories, with no overuse of virtual observations for the longest trajectories.

### 2.3 A similarity coefficient to compare atlases

With the development of multiple detection and tracking algorithms, the eddy community lacks a tool able to
provide quantitative and qualitative comparison of the atlases. The similarity coefficient provides the association of detected and tracked eddies between two (or more) atlases, with the quantification of the similarity of their respective physical characteristics.

#### 2.3.1 Principle

The similarity coefficient (SC) compares the eddies detected in different databases having the same formalism :
each eddy is saved with a rotating sense, position, amplitude, radius, time, trajectory, and contours. As for the tracking procedure, when we want to associate similar eddies within a trajectory, here we search for one eddy at a time $t$ in one atlas (the reference) and check if there is a corresponding eddy in another atlas (the study) at the same time, using the overlap of the effective contours of those eddies. The similarity coefficient is defined in Eq. (1) as the ratio between the intersection of the eddies' effective areas in the reference and the studied atlases
and the union of their effective areas, expressed in percent :

$$Similarity\ Coefficient = 100 \times \frac{Area\left(Eddy_{ref}\right) \cap Area\left(Eddy_{study}\right)}{Area\left(Eddy_{ref}\right) \cup Area\left(Eddy_{study}\right)} \tag{1}$$

Here, we will investigate the SC computed for anticyclones and cyclones separately, but the SC can also be
performed by cross-referencing anticyclones and cyclones, in order to evaluate the occurrence of opposite detection in the atlases.

When an eddy has no match in the other atlas, its similarity coefficient is 0 %. When the two eddies are identical (same contour and position), their similarity coefficient is 100 %. We define four specific groups of eddies depending on their SCs : *i)* unmatched eddies, with no association or very low overlap (SC < 5 %) ; *ii)* different
eddies (eddies with low SCs, between 5 and 20 %) ; *iii)* intermediate eddies (eddies with SCs between 20 and 40 %) ; *iv)* similar eddies (eddies with high SCs, over 40 %) (Figure 3). Eddies can be well positioned but with different radii: in the idealized case of two eddies represented by circles where one is included in the other, a SC above 40 % implies a maximum ratio between the eddies' radii of $\sqrt{40}$ % ≈ 0.63, thus very similar eddies in location but also physical characteristics. Eddies with no similarity coefficient (0 %) are the novelty, as they
were not present in one atlas. Eddies with low SCs are representative of the differences between the two atlases, as they are present in both but quite different in their location and characteristics. Note that with the SC definition used here, an eddy included in another eddy or shifted eddies are treated similarly (Figure 3).

Since the META2.0 product does not provide eddy contours, in all comparisons made with this dataset we replaced contours by the circles built from the center and the radius values.

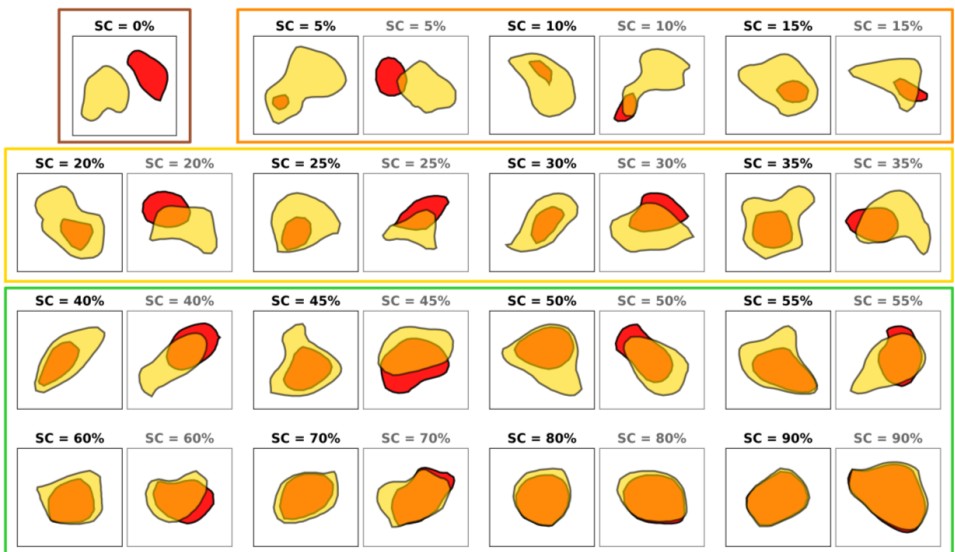

**Figure 3 : Illustration of the Similarity Coefficients (SCs) from 0 % to 100 %. A SC of 0 % is when the eddies of one atlas have no match in the other atlas (brown rectangle), SCs strictly below 20 % are associated with quite different eddies (orange rectangle), SCs between 20 – 40 % are considered as intermediate (yellow rectangle), and SCs over 40 % are associated with very similar eddies (green rectangle). Except for unmatched eddies (SC = 0 %), for each SC is presented on the left the case of one eddy included in the other, and on the right the case of shifted eddies.**

After the visualization of the SCs between different atlases, we noticed that some complex oceanic regions (high latitudes where sea ice can be present, semi-enclosed seas with complex dynamics and topography) have lower SCs and more unmatched eddies than in the open ocean, thus we decided to restrict the similarity coefficient applications to the major open oceanic basins. Only ~10 % of eddies are removed from the analysis with this open ocean selection. These non-selected eddies are available in both META2.0 and META3.1exp, users interested in these critical zones are welcome to analyze them regionally and provide feedbacks. The following results are all obtained with this geographic mask.

### 2.3.2 Validation of the similarity coefficient capacity : study case of the all- versus two-satellites detection

To evaluate the ability of the similarity coefficient in depicting changes with accuracy, we present here the results obtained with the following datasets. The reference atlas is the detections made with PET on two-satellites ADT maps and the study atlas is made with PET on the all-satellites ADT maps. Both atlases use the high-pass filter with a 700 km cutoff wavelength. The only difference between the atlases is the number of satellites used in the production of the SSH maps and the way the along-track SLA is built (see Sect. 2.1).

The similarity coefficient captures well the influence of the temporal variation of the satellite constellation in the representation of mesoscale eddies (Figure 4). From 1993 to 2000, the period where only two altimetric missions provided data, 94.3 % of the detected eddies of the all-satellite based atlas have SCs higher than 40 % (very similar eddies). The new eddies (SC = 0 %, red) represent 3.6 % of the dataset. Eddies with the lowest



SCs (5 – 40 %) represent only 1.9 % of the dataset, and the 0.2 % multiple matched eddies are anecdotal. The
small differences between eddies detected in the all- and two-satellites products are directly linked to the
different processing used to build the maps, with the absence of mean profiles in the two-satellite product. The
SCs are also very homogeneous in time, except from January 1994 to March 1995 where only Topex/Poseidon
was delivering data.

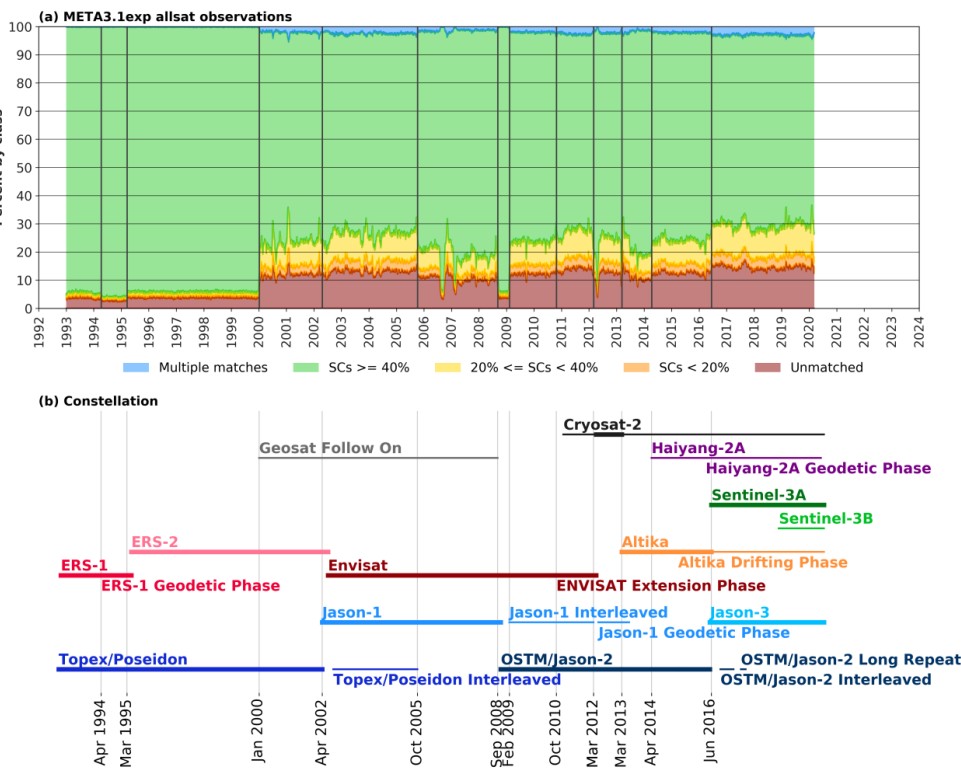


**Figure 4 : (a) Temporal evolution of the similarity coefficient distribution obtained for the META3.1exp all-satellites,
compared with the META3.1exp two-satellites. The percentage of unmatched observations (dark red shading)
represents eddies detected in the all-satellite product but absent in the two-satellite atlas. The low and intermediate
similarity coefficients (SCs < 40%, yellow and orange shadings) highlight eddies with shifted locations. Eddies**
**similarly detected in the all-satellites and two-satellites products represent most of the eddies (SCs ≥ 40%, green
shading). (b) Temporal evolution of the satellite constellation used for the two-satellites (thick lines, reference
missions) and the all-satellite (thick and thin lines) products. The highlighted changes in the missions correspond to
some abrupt variations in the SCs.**

After 2000, not only the processing of the maps is different, but the number of satellites changes, with up to 6
satellites in the constellation. The introduction of more than two satellites is well captured by the similarity
coefficient method, as the rupture in the homogeneous spatio-temporal two-satellites sampling coincides with a
reduction of the observations with the highest similarities between the two-satellites and the all-satellites
detection (only 73.7 % of observations with SCs ≥ 40 %). Multiple matched eddies are present in few but higher
quantities (2.2 %). The proportion of the lowest SCs is 11.7% in mean, and the new eddies represent also 12.4%



of the all-satellites dataset. The repartition of the SCs in time is not homogeneous but reacts to the changes in the constellation (Figure 4), highlighting the sensibility of the eddy detection to the mapping procedure. The geographic distribution of the SCs after 2000 (not shown) indicate strong similarities under the reference mission tracks, which is expected as the reference missions are shared by the all- and two-satellites products,

and lower similarities and new eddies in the diamond-shaped observation gaps between the reference tracks, where the additional satellites enhance the mesoscale representation. Thus, the increased number of eddies with the lowest similarity coefficients testifies for a repositioning of the mesoscale structures. The new eddies are captured by the higher spatial resolution of the satellite tracks in the all-satellite product.

**3 Results**

**3.1 Assessment of the PET algorithm and parameters**

We recall that in addition of the modification of the tracking scheme, three major changes are made from the META2.0 to the META3.1exp version : 1) the change of input maps, from SLA to ADT; 2) the change in the filtering step, from 1000 km to 700 km for the cutoff wavelength of the 2D filter; 3) the detection algorithm evolution, from OSU to PET. These changes need careful evaluation, not just a simple analysis of the

distribution of the main characteristics of the detected eddies in the final META2.0 and META3.1exp products, even with the similarity coefficient. With a direct comparison, the impact of the various changes on the eddies' main characteristics will be mixed and could compensate each other. Thus, we produced intermediated atlases to assess separately each change.

For each comparison step, we characterized the continuity between the tested datasets, the novelty, and their

differences.

**3.1.1 Eddy detection and tracking from SLA to ADT**

Historically, the detection of eddies was made on SLA maps, whatever the detection method (Chaigneau et al., 2009, 2008; Chelton et al., 2011, 2007; Dilmahamod et al., 2018; Dong et al., 2014; Mason et al., 2014; Morrow et al., 2004; Yi et al., 2014), mainly because the SLA maps were the most reliable altimetric field, due to

residual geoid errors in the mean. The collocation of in situ data with detected eddies confirmed the robustness of SLA-based detections in the open ocean, with a clear distinction between AEs and CEs in agreement with the theory (Castelao, 2014; Chaigneau et al., 2011; Keppler et al., 2018; Melnichenko et al., 2017; Pegliasco et al., 2015; Zhang et al., 2016; Zu et al., 2019). Nevertheless, those studies were conducted in open-ocean regions where the MDT is mostly homogeneous, thus the SLA was able to represent correctly the mesoscale eddies

following their basic description : rotating structures in a homogeneous background, which appear as anomalies. But in specific areas where the mean circulation has strong spatial gradients, various studies highlighted the discrepancies between in situ observations and SLA-detected eddies, such as in the Mozambique Channel (de Ruijter et al., 2002; Halo et al., 2014; Schouten et al., 2003). The SLA, as stated by its name, is an anomaly over a temporal mean. But when the temporal mean contains the signature of a mean mesoscale structure, such as a

recurrent meander or eddy, the SLA only reflects the variation of the sea surface height relative to the mesoscale structure (Rio et al., 2014). Thus, a positive SLA can either represent an anticyclonic eddy, the weakening of a cyclonic meander or eddy, or the reinforcement of an anticyclonic meander or eddy. Similarly, a negative SLA



can be a cyclonic eddy, reflect the weakening of an anticyclonic circulation or the reinforcement of a cyclonic circulation.

With the more recent improvements in the quality of the MDT (Rio et al., 2014), several studies started being based on ADT-detected eddies, in the Agulhas Retroflection region (Doglioli et al., 2007; Laxenaire et al., 2019, 2018; Rubio et al., 2009) or the Mediterranean Sea (Ioannou et al., 2017; Le Vu et al., 2018; Mkhinini et al., 2014), both regions being known for their non-homogeneous mean circulation. In the Mediterranean Sea, the comparison of SLA- and ADT-detected eddies led to the conclusion that ADT fields should be preferred over

SLA fields when detecting eddies in oceanic regions with inhomogeneous MDT (Pegliasco et al., 2020). In the global ocean, the regions where the MDT contains sharp gradients or recurrent meanders are the most energetic regions where strong currents are present, such as the Antarctic Circumpolar Current, the Kuroshio, the Gulf Stream, the Agulhas Retroflection, and the Zapiola Eddy. These regions play also an important role for the redistribution of heat over large scales, involving the eddies generated by the destabilization of those strong

currents (Sun et al., 2019). An accurate detection of eddies in those specific areas is thus of great importance. The similarity coefficient was applied to compare the intermediate atlases of the open ocean trajectories lasting at least 10 days, with a detection made with the PET algorithm on the two-satellites ADT (59.4 million detected eddies) and SLA (61.7 million detected eddies) fields, both filtered with a 700 km cutoff wavelength.

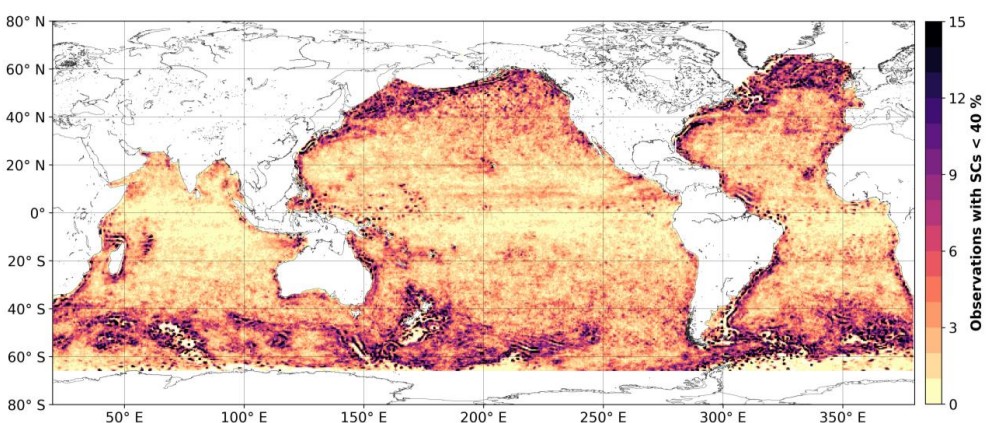

**Figure 5 : Number of eddy centers associated with ADT-detected eddies that are unmatched or matched with low similarities indexes (SCs < 40 %) with SLA-detected eddies.**

77 % of the eddies detected and tracked more than 10 days in the ADT maps matched with SCs higher than 40 % the eddies detected and tracked on SLA maps. So the majority of open-ocean eddies have a similar distribution of radii, amplitude, and lifetimes, whether we use the original ADT or SLA fields. Of interest here

are the smaller group of 11 % of the ADT-detected eddies that are newly detected, as well as eddies with SCs below 40%. Less than 1% of eddies have multiples matches.

The lowest SCs and the new eddies are concentrated in the coastal regions, near islands, and in the strong open-ocean current systems mentioned above (Figure 5). These are often regions where the filtered MDT contains specific patterns (Figure 7). Since closed MDT circulations can be present in the ADT maps but not in the SLA

maps, new eddies with the same rotating sense are generated in those closed MDT circulations. This implies a



strong polarization of the presence of the ADT-detected cyclones and anticyclones, whereas eddies are more homogeneous in space in the SLA maps, regardless of their rotation. Figure 6 highlights the difference in the percentage of time spent by each grid point within cyclones or anticyclones, in the northwestern Atlantic, including the Gulf of Mexico and the Gulf Stream, two regions where strong differences arise depending on the

SSH used for the detection.

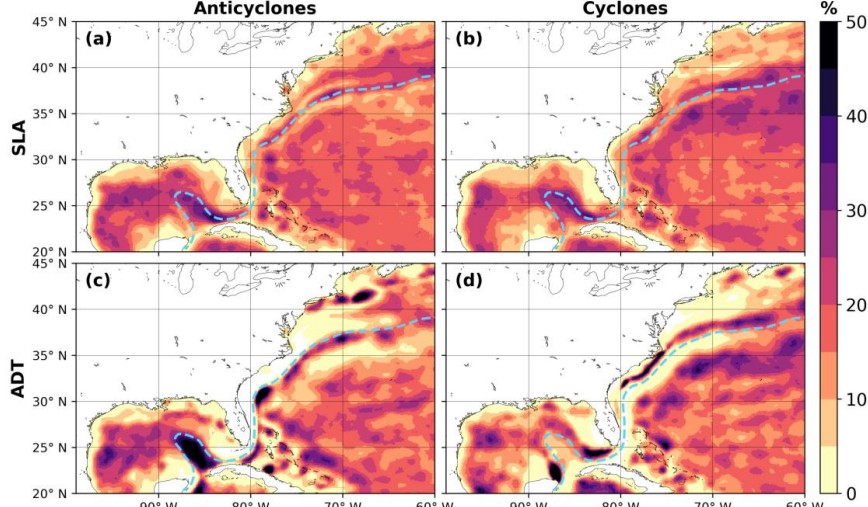

**Figure 6: Percentage of time spend by each pixel of 0.1°x0.1° within anticyclones detected in (a) SLA and (c) ADT maps and within cyclones detected in (b) SLA and (d) ADT maps. The dashed blue line follows the velocity maximum of the current.**


In the Gulf of Mexico, the main Loop Current enters the Gulf from the south via the Yucatan Channel and circulates as a loop (dashed blue line in Figure 6). An anticyclonic eddy sometimes detaches from the loop and drifts through the Gulf, named the Loop Current Eddy (LCE). This specific circulation is associated with Frontal Loop Current Cyclonic Eddies (FLCEs) on the edge of the current, that can participate in the shedding of the

LCE by pinching the current (Le Hénaff et al., 2012; Schmitz, 2005). The ADT-detected eddies collocate well with these anticyclonic and cyclonic circulation features, whereas the SLA-detected eddies are more homogeneous in the Gulf of Mexico, with cyclones as frequent as anticyclones at the mean position of the Loop Current. Rather than being of specific polarity, these SLA eddies represent the waxing and waning of the Loop Current and the Loop Current Eddies.

Eddies generated from the Gulf Stream vein are also quite different depending on their detection on SLA or ADT maps. The SLA-detected anticyclones and cyclones are both often present within the principal vein of the current (Figure 6a, b, and dashed blue line), with fewer occurrences immediately south of it. Those eddies are often surrounded by eddies rotating in the opposite sense, thus preferentially cold-core cyclones to the south and warm-core anticyclones to the north. For the ADT-detected eddies, the north-south difference is clearer, with a

majority of anticyclones detected just south of the mean axis, and cyclones dominating to the north (Figure 6c,d). ADT-detected anticyclones are found again at higher latitudes, and cyclones below ~35°N. Note that





these high occurrences away from the main vein may be related to the large rings initially embedded in the meanders and known for their clear signature in SST images, whereas the marked occurrences near the vein are likely linked to the continuous small recirculations on both sides of the current (Chi et al., 2019; Liu et al., 2018;

Waterman and Hoskins, 2013). Such behavior occurs similarly in other oceanic regions where the MDT contains sharp gradients. For those regions, SLA-detected structures can reflect the reinforcement, weakening or displacement of mean jets and eddies with an imprint in the MDT, instead of the long-term polarity of the structures. Users specifically interested in such areas are welcome to provide feedbacks.

### 3.1.2 Adaptation of the filtering step

In the first census of detected and tracked eddies at global scale and for the whole altimetry area, Chelton et al. (2011) implemented a pre-processing step to filter out the large scale from the SLA maps. This filter of 20° longitude and 10° latitude half-power cutoff wavelengths efficiently removed the large SLA scales such as the heating and cooling steric effects and other large-scale patterns, like the chevron patterns visible in the tropical Pacific (See Figure 1 of Chelton et al., 2011). In a smaller regional study, Mason et al. (2014) proposed a

different large-scale filtering with a Gaussian filter of radius of 10° longitude and 5° latitude, and obtained similar results in the Canary Island region. The filter used to preprocess the META2.0 maps is a 2D Lanczos filter with half-power cutoff wavelengths of 1000 km on both the zonal and meridional dimensions, to prevent the impact of the non-equivalence of degrees and kilometric distances with latitude. This filter was adapted for SLA maps, where the background gradients cover oceanic basins. But in the ADT maps, strong localized

gradients linked to the MDT are present, particularly in the intense current areas (the Antarctic Circumpolar Current, the Gulf Stream, the Kuroshio, the Agulhas Retroflection, and the Zapiola Eddy). Those currents and their meanders have typical mesoscale dimensions. Thus, the filter must separate efficiently the small and the large scales of the MDT and of the SLA to provide adequately filtered ADT maps. The change from 1000 km to a smaller cutoff wavelength was motivated by the behavior of the filter in the intense current areas. Since the

MDT gradients are sharp, the larger filter cutoff expanded the gradients after filtering, implying a loss of the physical content of the filtered maps. Different half-power cutoff wavelengths were tested, including no large-scale filtering, and SLA and ADT eddy detections were made from the various filtered maps. A compromise was reached, between keeping all of the mesoscale structures in the strong current areas of the MDT in the filtered maps, and properly filtering the large-scale equatorial patterns, by using a 700 km half-power cutoff

wavelength. With this filtering, we reduced the number of large-scale eddy-shaped structures that can be detected in the low frequency MDT and maintained the dimensions of the remaining eddy-shaped structured above 300 km radius. The remaining high pass filtered MDT structures are presented in Figure 7. The large-scale patterns in the equatorial band are efficiently removed and the strong currents gradients are preserved.

To ensure the continuity of the eddy representation with this change in large-scale filtering, we applied the

similarity coefficient to the trajectories obtained from the PET detection on the two-satellites ADT maps filtered with a half-power cutoff wavelength of 1000 km and 700 km. The number of the selected eddies is close, with 55.5 million detected eddies for the 1000 km half-power cutoff wavelength and 59.4 million detected eddies at 700 km. 87 % of the open ocean eddies detected and tracked for more than 10 days with the 700 km half-power cutoff wavelength matched those detected and tracked with a 1000 km half-power cutoff wavelength, with SCs

higher than 40 %. Only 9 % of them are new eddies, and 4 % have SCs below 40%. The lowest SCs and the new



eddies are concentrated in the strong current areas mentioned above, since the decrease in the half-power cutoff wavelength maintains more of the sharp gradients of these regions than with the original half-power 1000 km cutoff wavelength. The 700 km cutoff wavelength induces a slight increase of the amplitude and the radii for the eddies with high SCs detected in the 700 km filtered ADT maps compared to the eddies detected in the 1000 km
filtered ADT maps, but without influencing the lifetime of the built trajectories.

Thus, the change in the filtering step provides a continuity of the detected eddies between the previous and the new half-power cutoff wavelength, yet ensures an adapted representation of eddies in the strong current areas in ADT maps.

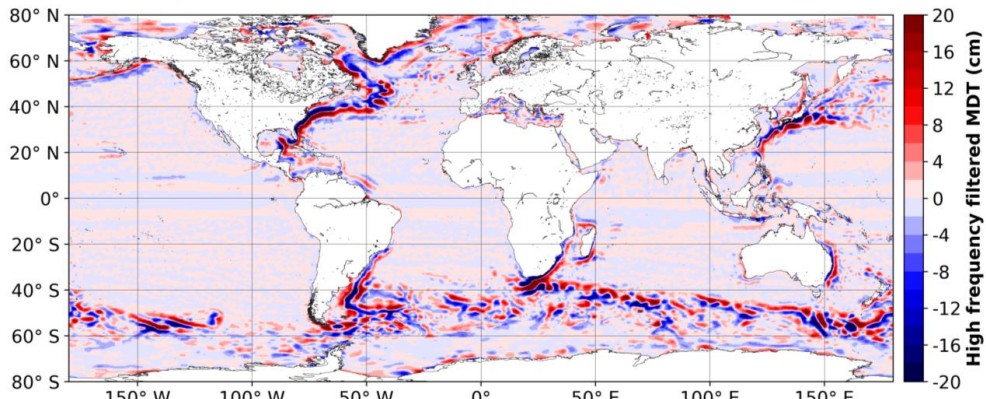

**Figure 7 : High frequency of the Mean Dynamic Topography filtered with a 700 km half-power cutoff wavelength.**


### 3.1.3 Change in the detection algorithm : from OSU to PET

The major change between the META2.0 and the META3.1exp datasets is the change in the detection algorithm. To evaluate the impact of a detection made with the PET algorithm instead of the historical OSU one, we produced an intermediate dataset similar to META2.0 except for the detection algorithm.

We applied the PET detection algorithm to SLA maps filtered with a 1000 km half-power cutoff wavelength, and tracked the detected eddies with the restricted area method. We lowered the minimum lifetime threshold to 10 days and provide statistics only for the open ocean. The first striking result is that even with minimum lifetimes of at least 10 days, the PET-detected eddies outnumbered the OSU-detected eddies by a factor of 1.7, with respectively 5936 daily eddies for PET (59.4 million over the total period) and 3445 daily eddies for OSU
(34.9 million over the total period). Table 2 presents the repartition of the detected eddies after the pairing between the datasets. Despite the large number of new eddies detected with PET (1994 daily new eddies), there is also a higher quantity of eddies with high SCs (2318 daily eddies), meaning there are a larger number of similar eddies detected between META2.0 and the intermediate atlas. In comparison, the low and intermediate SCs represent only 575 daily eddies. Contrary to the other analyses in this paper, PET eddies often have
multiple matches (17 % of the datasets) and a large number of unmatched eddies (33 %), and need to be investigated.




| Daily number of eddies | Total | High SCs | Intermediate SCs | Low SCs | Multiple matches | Unmatched eddies |
|---|---|---|---|---|---|---|
| OSU (META2.0) | 3445 | 2153 (62%) | 319 (9 %) | 237 (7 %) | 603 (17 %) | 133 (4 %) |
| PET | 5936 | 2318 (39 %) | 336 (6 %) | 239 (4 %) | 1049 (18 %) | 1994 (33 %) |

**Table 2 : Averaged daily number of eddies detected with the OSU or the PET detection algorithms and the percentages relative to the total in parenthesis. The atlas generated with the PET detection mimics the META2.0 parameters. Note that the eddies are part of trajectories with a minimum lifetime of 10 days.**

To qualitatively compare the eddies detected with the two algorithms, we analyzed the repartition of the SCs' categories as a function of the effective radius (Figure 8) and amplitude (not shown). To provide an idea of the distribution of the effective radius for each dataset, we represent for each SC's category its percentage relative to a constant number of eddies. The vertical lines of the Figure 8 delimit the bounds of each effective radius class containing 5 % of the total eddy population (~1.7 million eddies in a class for OSU, ~2.9 million eddies for PET). Lines close to each other highlight a concentration of eddies for the corresponding effective radii. In general, the PET-detected eddies' population has smaller effective radii than the OSU-detected eddies' population, with a median radius of 55 km for PET and 90 km for OSU. Nevertheless, the population of OSU-detected eddies matching the PET-detected eddies with high SCs is present for all the effective radius classes (Figure 8a, green), except for the smallest PET-detected eddies with effective radii below 30 km. 62 % of the OSU-detected eddies are thus detected similarly with the PET algorithm, that ensures the continuity between the two detections.

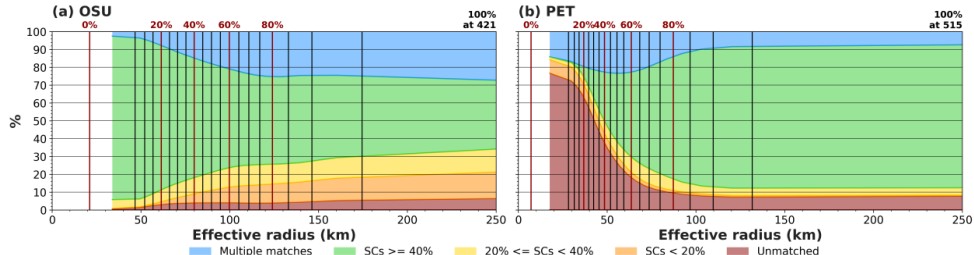

**Figure 8 : Similarity coefficient repartition as a function of the effective radius for the eddies detected with (a) OSU and (b) PET over the 1000 km filtered SLA. The vertical lines separate the eddies in 20 equally distributed classes of ~1.7 million eddies for OSU and ~2.8 million eddies for PET.**

The new eddies, represented by the unmatched PET-detected eddies, have a quantity comparable with the similarly detected eddies (33% of the PET dataset for the unmatched eddies, 39% for the high SCs eddies, Table 2). The new eddies are over-represented in the smallest effective radius classes (Figure 8b, dark red). These eddies that were not detected with OSU may arise from three OSU thresholds in the detection scheme : the absence of detection near the Equator (± 2.5°), the amplitude limitation at 1 cm, and the absence of gaps between pixels in latitude or longitude. The PET algorithm detects eddies near the equator, accepts amplitudes above 0.4 cm, and tolerates a maximum shape error of 70 %. With only 91 daily PET-detected eddies near the equator, the change in amplitude threshold is responsible for a third of the new eddies (671 daily eddies under 1 cm, 37 of them near the equator). The remaining two-thirds of new eddies (1269 daily new eddies) with



amplitudes higher than 1 cm, including 555 daily eddies with amplitude higher than 2 cm, are located mainly in the coastal areas, in the ACC, in the Irminger Sea and East of the Reykjanes Ridge. Note that the requirement of having no gap between adjacent pixels combined with the 1 cm amplitude threshold in the OSU algorithm might

block the eddies' detection near the coastline due to the presence of land pixels, whereas the PET algorithm, with the interpolation of the SSH levels, can more easily close contours in these areas.

Regardless of their size or amplitude, we are interested in the behavior of these new detected eddies from a dynamical point of view, since being part of a long trajectory is a synonym of persistency. We note that these new, unmatched PET-detected eddies are present homogeneously from 15 % to 85 % of the normalized

lifetimes, with a slightly higher amount present during the early phase (0 – 15 % of the normalized lifetime) and at the very end (85 – 100 % of the normalized lifetime) (not shown). Smaller eddies are expected to be particularly involved in the growth and the decay of the trajectories. This result reinforces the robustness of the new eddies, for the smallest as well as for the larger ones.

Eddies with multiple associations are particularly numerous when comparing the OSU and PET detections

(~17 % of the datasets). The multiple matches category is divided in three subgroups : the parents of twins, where one eddy matches with two eddies; the twins, associated with a unique parent eddy; and the complex multiple matches, where more than two parents or children associations coexist. This last subgroup represents only ~2 % of the datasets. It is more frequent to associate one larger eddy detected with OSU with two smaller eddies detected with PET than the reverse : 12 % of the OSU-detected eddies are parents, and 3 % are twins,

whereas 1 % of PET-detected eddies are parents and 15 % are twins. This is directly linked to the possibility in the OSU detection to have more than one extremum within an eddy. This specificity was initially developed to treat the SSH irregularities or to detect eddies in close proximity, and avoid too large changes in the eddy's characteristics from one weekly map to another (Chelton et al., 2011). With the improved altimetric maps available daily nowadays, detecting eddies with more than one extremum often results in the association of two

unrelated eddies, with clear separated trajectories, but in close proximity. The OSU multiple extrema eddies also tend to have large effective radii and amplitudes compare to the corresponding unique PET-detected eddies. This implies that when the multiple extrema structure separates in two (or more, but rarely) eddies, the change in radius and amplitude is quite important, and thus exceeds the limitations specified in the restricted area tracking. Thus, instead of following one of the two (or more) structures, the trajectory is stopped and a new one

starts for each separated eddy.

One example of the complex associations of eddies in a double extrema eddy detected by OSU is highlighted in Figure 9. One single META2.0 main dark green trajectory is followed along its path west of Australia for more than 6 months. The PET detected eddies are associated in three smaller but coherent trajectories (PET1, PET2, PET3, see Figure 9). At first, PET1 (in yellow) is coherent with the beginning of the META2.0 investigated

trajectory. On the 22 December 2008 (Figure 9b), PET1 and PET2 (in red) are close enough to be detected as a single multiple extrema eddy in the META2.0 investigated trajectory. Nevertheless, the two independent PET trajectories do not merge and remain separated, as visible in the SLA contours and the chlorophyll background in Figure 9b and Figure 9c. Between the 15 January 2009 (Figure 9c) and the 26 January 2009 (Figure 9d), the META2.0 detection finally stops considering a large structure including PET1 and PET2 and only follows

PET2. For this step, the restricted area-tracking scheme of META2.0 includes a virtual observation corresponding to an isolated position, due to the very rapid shape change. Meanwhile, the PET1 trajectory was



caught by another META2.0 trajectory (in lime green). A similar scenario is repeated after the 9 February 2009 (Figure 9e) with the investigated trajectory encompassing the PET2 and PET3 (in orange) trajectories from the 23 February to the 20 April 2009 (Figure 9f,g,h,i). Allowing the detection of a multiple extrema eddy led to a

jump in the eddy's position, as it is calculated from the centroid of the SLA within the eddy. But again, despite the proximity of the two PET trajectories, the SLA and the chlorophyll maps attest to the absence of intense eddy interactions occurring before the main merging events. A few days after the 20 April, the META2.0 investigated trajectory is stopped due to an important variation in the position and the radius of the eddies. A new META2.0 trajectory is started, following the PET2 trajectory (lime green, Figure 9j) whereas the PET3

trajectory corresponds to another META2.0 trajectory a few days before the 30 May 2009 (Figure 9k). Even though the main path of the trajectories depicted here is quite similar between PET and META2.0, their evolution is very different. The PET eddy trajectories never interacted enough to be considered as unique structures, as showed by their independent chlorophyll signature, and PET2 and PET3 trajectories last longer than the META2.0 trajectory, without interacting with PET1. The possibility of detecting multiple extrema

eddies in META2.0 may be an anticipation of merging events, or a better representation of splitting events, but they are also more likely to merge independent eddies. In addition to the inconsistency of the META2.0 trajectory dynamics, the META2.0 representation of eddies as an isotropic circle, and not by the real eddy contours, can lead to wrong associations of external data with the eddies, in particular in the case of the multiple extrema eddies. The series of snapshots (Figure 9b-k) highlight that the area encompassed by the circles are

likely to mix signatures from the eddies and the background field, and from different but geographically close eddies, being of the same rotation or not. This example is illustrated online (https://www.youtube.com/watch?v=4Vs3ZJNMViw). In the video, we released particles within the eddies' contours on the 11 December 2008, and then advected them over time for the PET trajectories and the META2.0 investigated trajectory. The particles stayed within the PET eddies' cores much more than in the

META2.0 investigated trajectory and the absence of merging events is highlighted, as each PET trajectory keep most of its particles within its eddy's contours. The particles clearly showed the benefit of providing eddy's contours instead of circles, and of detecting unique instead of multiple extrema eddies in order to represent properly the mesoscale eddies' dynamics.

This example is not an isolated case, the behavior of META2.0 trajectories, especially involving multiple cores

eddies, can be quite different from PET built trajectories. Unfortunately, metrics to quantify exactly how they differ are not available due to the complexity of the problem.

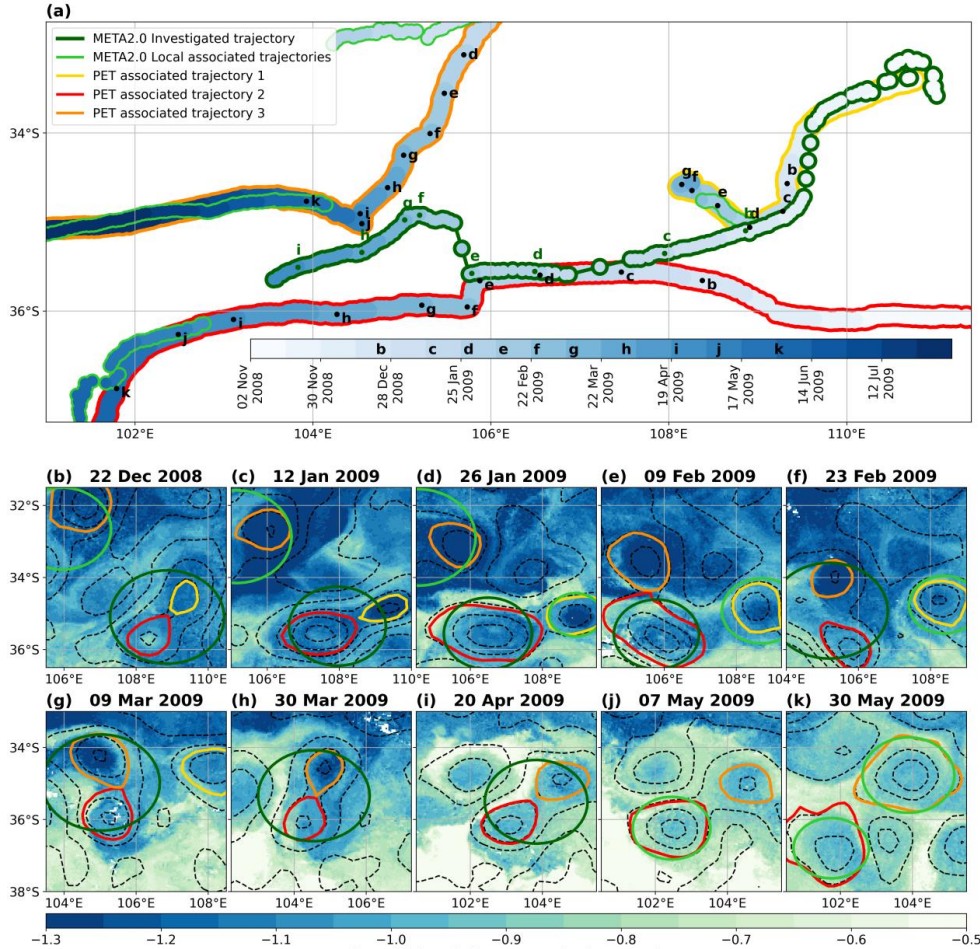

**Figure 9 : Selected eddies detected by both OSU (META2.0) and PET algorithms. The input field (filtered SLA) is identical for both detections, as the tracking (restricted area method). Eddies were selected as representative examples highlighting the differences between the two detection algorithms. Panel (a) shows six META2.0 trajectories (the dark green investigated trajectory, and five smaller ones in light green) and three PET trajectories (yellow, red and orange) for eddies detected over the same temporal period, from November 2008 to January 2009 (color scale). Dashed black lines in panels (b-k) show arbitrary 10 cm levels of filtered SLA. Background colors in panels (b-k) show chlorophyll-a concentrations in logarithmic scale (data courtesy of CLS). Note the reverse color scale, showing high concentrations in light color. Letters on panel (a) correspond to the snapshot time of panels (b-k) as well as panel (a) trajectories outline colors correspond to contours in panels (b-k).**

## 3.2 Assessment of META3.1exp product

The new META3.1exp product differs from the META2.0 product for the input SSH field, the half-power cutoff wavelength used to filter it, the detection and the tracking algorithms. We saw in the Sect. 3.1. that the change of the input field from SLA to ADT slightly decreased the number of detected eddies, whereas the change in the cutoff wavelength from 1000 km to 700 km increased the number of detected eddies, but neither changes





impacted strongly the radii and amplitudes' distributions. On the other hand, the PET algorithm detected many
        more eddies than the OSU algorithm on the same maps (1.7 times more), and the change from the restricted area
        method to the overlap tracking increased also the number of trajectories lasting more than 10 days, especially
        for the longest lifetimes. Thus, when comparing directly the META2.0 and META3.1exp number of eddies and
        trajectories (Table 3), we can attribute the higher number of eddies in META3.1exp (1.8 times more) to the
change of the algorithm. This implies more built trajectories, and the change in tracking increases the percentage
        of trajectories lasting more than 30 days in the META3.1exp product compared to the META2.0 product. The
        number of eddies involved in the longer trajectories is similar in percentage between the two products, but
        represent almost twice as many eddies in META3.1exp than in META2.0. Note that the short trajectories (10 to
        30 days) of META2.0 were not delivered to the users, but were processed here to be coherent with the
META3.1exp description.

| Lifetime (days) | | [10, 30[ | [30, 90[ | [90, 180[ | [180, 270[ | [270, 360[ | ≥ 360 | Total |
|---|---|---|---|---|---|---|---|---|
| META2.0 | Eddies | 24.6% | 37.0% | 21.0% | 8.5% | 4.1% | 4.7% | 35,700,819 |
| | Trajectories | 60.0% | 30.5% | 7.0% | 1.6% | 0.5% | 0.4% | 878,777 |
| META3.1 | Eddies | 20.1% | 36.9% | 23.6% | 9.5% | 4.2% | 4.8% | 64,113,623 |
| exp | Trajectories | 55.4% | 33.0% | 8.6% | 2.0% | 0.6% | 0.4% | 1,433,336 |

**Table 3 : Number of individual eddies and trajectories for different lifetimes in the META2.0 and the META3.1exp products. Note that the online META2.0 product only contains eddies associated with trajectories lasting more than 28 days.**

As noted in the previous section, the difficulty in comparing mesoscale eddies atlases is to identify which eddies
        are new, and in the conserved eddies, if they are similar or not. We provide in Figure 10 the global distribution
        for the effective radius and the amplitude of eddies detected and tracked for at least 10 days in the META2.0
        and the META3.1exp product. Remember that 62 % of the META2.0 eddies are associated with high SCs with
        the PET-detected eddies over the same input field (SLA filtered with a 1000 km half-power cutoff wavelength)
and that 33% of the PET-detected eddies are new, with a high proportion of small size eddies, both in amplitude
        and radius. The shorter trajectories lasting from 10 to 30 days have more than 5 % of the involved eddies with
        effective radii below 50 km in the META2.0 product, whereas a large number of eddies with these small
        dimensions are part of the trajectories of META3.1exp, whatever their lifetime (Figure 10a). Indeed, a large
        number of eddies with amplitudes below 1 cm that are detected in META3.1exp (but not in META2.0) are
associated with short trajectories, but there are still small-amplitude eddies present in the longer trajectories
        (Figure 10b), especially at the start and the end of the trajectories. It is clear that the smaller structures have a
        physical consistency since they contribute to the short and the long trajectories, despite being close to the
        resolution of the altimetric maps.

        For the larger eddies, the amplitude values associated with the 75th percentile for the different lifetimes is always
lower in the META3.1exp product than in the META2.0 product, except for the trajectories lasting more than
        one year. For the effective radius, the median values of the isotropic META2.0 eddies are higher than the
        META3.1exp more complex eddies by 35 km, for all lifetimes higher than 30 days. The largest (95th) percentile
        of the effective radius in META2.0 decreases as the lifetime increases, whereas in META3.1exp this value
        increases to reach a plateau around 145 km. These shifts for the higher radii and amplitudes can be explained by

the absence of multiple extrema within an eddy in META3.1exp, since the presence of those multiple extrema eddies are a source of discrepancies in the construction of coherent trajectories, and poorly linked to the chlorophyll data (Figure 9).

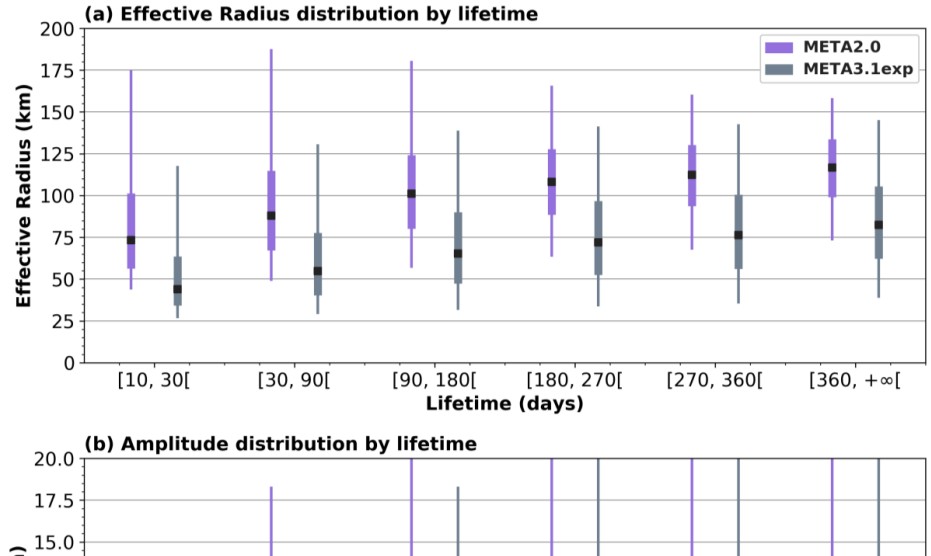

**Figure 10: Distribution of (a) Effective Radius and (b) Amplitude for different lifetimes for the META2.0 eddies (purple) and META3.1exp eddies (grey). The black squares mark the median, the thick lines span from the 25th to the 75th percentiles, and the thin lines from the 5th to the 95th percentiles of the data.**

Finally, following the analyses of Chelton et al. (2011), Figure 11 shows the different geographical distributions of the long-lived trajectories (more than a year) for META2.0 and META3.1exp. The number of trajectories in META3.1exp is twice that of META2.0, and the higher latitudes exhibit more trajectories. The main
geographical differences are linked to the increased detection from using ADT maps and not SLA. In particular, the more energetic regions are highly variable in SLA allowing the detection of anticyclones and cyclones over mean patterns, with very different patterns in the ADT detection. As presented in Sect. 3.1.1, the ADT built trajectories are organized following the MDT patterns. The META3.1exp shows more dynamical trajectories than META2.0 when tracking the anticyclonic Agulhas Rings and the cyclonic center of the Zapiola Eddy in the
South Atlantic, the anticyclones and cyclones generated in the lee of the Cape Verde or the Hawaii islands, the cyclones generated by the Leeuwin Current West of Australia and anticyclones generated South, the southern part of the Kuroshio and Gulf Stream veins, the Haida anticyclones in the Alaskan Gyre, and all the recurrent

anticyclones of the Mediterranean Sea. Nevertheless, some very static trajectories at high latitudes are suspected to be linked to the mapping methodology and thus should be taken with caution, and warrant further validation

through independent data colocation.

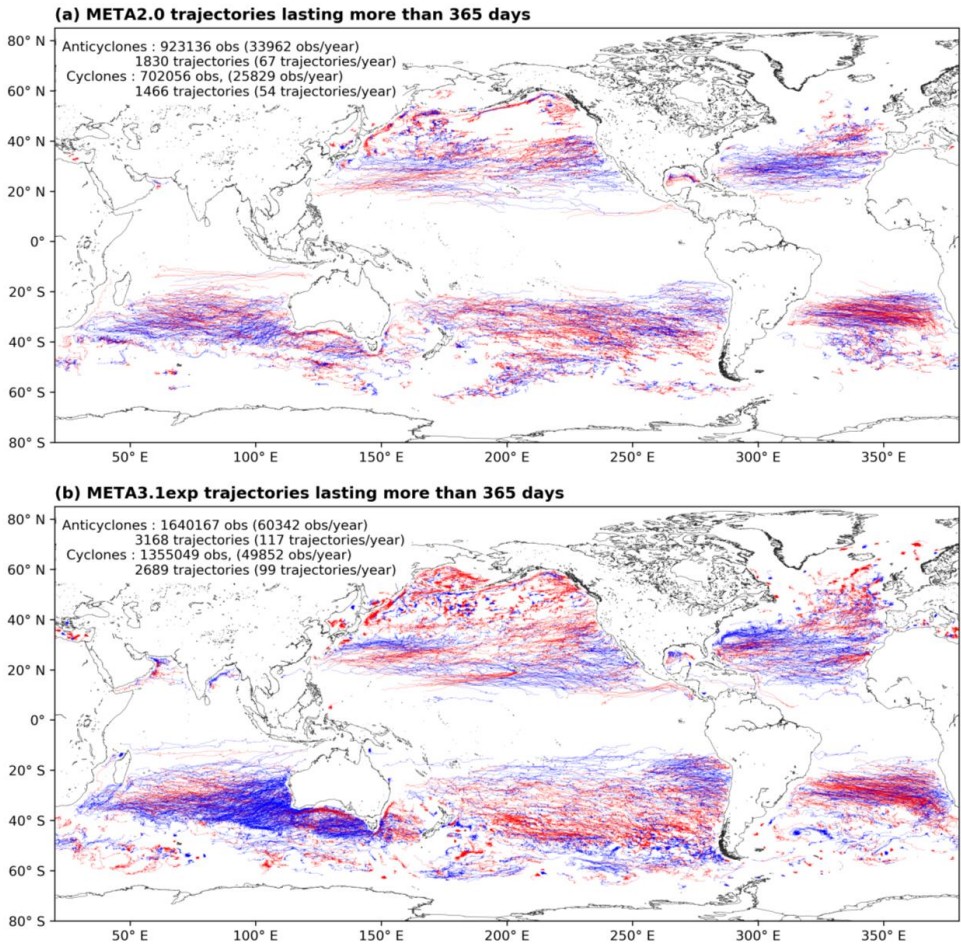

**Figure 11 : Trajectories lasting more than 365 days from (a) META2.0 and (b) META3.1exp. Cyclones are in blue, anticyclones in red.**

**4 Code and datasets availability**

The detection and tracking algorithms, as well as the implemented similarity coefficient, are freely released under a GPL V3 license (https://github.com/AntSimi/py-eddy-tracker) in the Python language. A large gallery of illustrated and documented routines to help with the data manipulation and visualization is available, and is updated when new methods are developed (https://py-eddy-tracker.readthedocs.io/en/v3.3.1/).



We produced two versions of META3.1exp in Delayed Time. The META3.1exp_twosat (https://doi.org/10.24400/527896/a01-2021.002, Pegliasco et al., 2021b) presented here is based on the two-satellites maps (C3S), ensuring a stability of the represented scales in space and time. This product is recommended for long-term analyses, including climate studies. The META3.1exp_allsat (https://doi.org/10.24400/527896/a01-2021.001, Pegliasco et al., 2021a) is based on the CMEMS all-satellites maps to take advantage of all the constellation, providing a better but inhomogeneous representation of the smaller scales in space and time depending on the available satellites (Figure 4).

The META3.1exp two-satellites and all-satellites products are available without restrictions on this AVISO repository : https://data.aviso.altimetry.fr/aviso-gateway/data/. The associated handbook describes the variables stored in the NETcdf files (SALP-MU-P-EA-23489-CLS, 2021). Six files are provided for each META3.1exp (Table 4). For both cyclones and anticyclones, the untracked files contain all the individual eddies with no association in trajectories, the "short" files are for the trajectories lasting strictly less than the minimum lifetime parameter, set here at 10 days, and the "long" files are for the trajectories lasting at least the minimum lifetime parameter. The global attributes of each file inform the users of the algorithm version used to product them, with their specific detection and tracking parameters. Be aware that the files are compressed, but loading them to use them will decompress the files. Since the contours are the more memory consuming, we recommend loading the files without these variables and make an extraction of the time period and geographic area of interest with the EddySubSetter application.

| Name of the file | Lifetime selection |
| --- | --- |
| META3.1exp_DT_twosat_Anticyclonic_long_%Y%m%d_%Y%m%d.nc | Trajectories lasting at least 10 days |
| META3.1exp_DT_twosat_Cyclonic_long_%Y%m%d_%Y%m%d.nc | |
| META3.1exp_DT_twosat_Anticyclonic_short_%Y%m%d_%Y%m%d.nc | Trajectories lasting strictly less than 10 days |
| META3.1exp_DT_twosat_Cyclonic_short_%Y%m%d_%Y%m%d.nc | |
| META3.1exp_DT_twosat_Anticyclonic_untracked_%Y%m%d_%Y%m%d.nc | Isolated detected eddies |
| META3.1exp_DT_twosat_Cyclonic_untracked_%Y%m%d_%Y%m%d.nc | |

**Table 4 : Nomenclature of the META3.1exp DT two-satellites NetCDF files.**

### 5 Summary

This paper aims to present the new global Mesoscale Eddy Trajectories Atlas (META3.1exp). Numerous evolutions have been made from the META2.0 product, mainly to provide useful characteristics for users, in particular the eddy contours, which are mandatory for efficient colocation of eddies with other data. Moreover, some details of the detection and tracking algorithm were developed for weekly altimetric maps, with less accuracy than the daily products available nowadays, and needed to be adapted. The code is freely released under a GPL V3 license (https://github.com/AntSimi/py-eddy-tracker) in the Python language and routines for data manipulation and visualization are documented.

The changes between the META2.0 and META3.1exp concern the input Sea Level field, its filtering, the detection algorithm, and the tracking scheme. We developed a methodology to compare intermediate datasets where only one change is made at a time, and to compute a similarity coefficient between matching eddies, from



the ratio between their overlap area and their union. Using these diagnostics, we have quantified eddies which are similar in the intermediate datasets (similarity coefficient over 40 %), those who differ (similarity coefficient between 5 – 40 %), eddies with multiples matches and the new eddies present in only one dataset. The analysis was made on restricted datasets including only eddies in the open ocean involved in trajectories lasting at least 10 days, although all geographical regions and shorter eddy trajectories are included in the full dataset.

Performing the detection of mesoscale eddies on the ADT instead of SLA maps allows us to better represent the ocean dynamics, since the SLA detected over strong MDT features represents their variability (weakening, reinforcement, displacement) but not their absolute polarity (anticyclones or cyclones). This eddy-detection change isolated from the others slightly reduced the number of detected eddies. Nevertheless, a large majority of eddies (77 %) are very similar between the ADT and the SLA detection in the open ocean, with no marked

influence on the radius and amplitude distributions. The major impact induced by the use of ADT maps is the geographical reorganization of the trajectories, with preferential occurrence of anticyclones (respectively, cyclones) over anticyclonic (respectively, cyclonic) MDT patterns in the META3.1exp dataset. To take into account the sharp gradients introduced in the ADT by the MDT while maintaining the mesoscale patterns in the filtered maps, the half-power cutoff wavelength was reduced from 1000 km to 700 km. This change increased

slightly the number of detected eddies, and the similar eddies (87 % of the dataset) had their radius and amplitude increased, but not significantly. The tracking scheme was also changed to improve the trajectories' reliability. Instead of searching for the eddy candidates to associate with the trajectory over a restricted area, the overlap method needs an overlap between the eddy and the next candidates, and the larger overlap is associated with the main trajectory in the case of multiple candidates. The number of consecutive virtual eddies introduced

to respond to the "missing eddy problem" was increased from three to four. The overlap method follows eddies over longer lifetimes, and this occurs without overuse of the virtual observations on the constitution of trajectories.

The major change comes with the detection algorithm change, from OSU, the historical detection algorithm, to the PET algorithm used to build the META3.1exp product. The PET algorithm detection is responsible for

almost doubling the detected eddies compared with META2.0. Nevertheless, more than 60 % of the META2.0 eddies are similarly represented in the PET detection. Among the different eddies, two specific populations have been identified. First, eddies not detected in META2.0 are involved in trajectories regardless of their size. Despite a non-negligible quantity of those eddies having small amplitudes, their time consistency prove they are not noise or artifacts due to the mapping procedure. The other interesting category is the multiple extrema

eddies authorized in the META2.0, whereas the PET algorithm only retains eddies with a unique extremum. This implies that 17 % of the META2.0 eddies have multiple extrema and are associated with two individual eddies detected with the PET algorithm. Despite their smaller sizes, the unique PET-detected eddies have more consistent lifetimes than the multiple extrema eddies. Note that the multiple extrema eddies do not correspond with individual eddies interacting before merging or after splitting events. Moreover, multiple extrema eddies

show less concordance with the chlorophyll data than the associated PET eddies. The combination of new eddies and two eddies instead of one multiple cores eddy in the META3.1exp product pull the amplitude and radius' distributions towards smaller values compared with META2.0, despite the consistency of effective radius and amplitudes for the similar eddies. The additional eddies in META3.1exp were not detected and



available at global scale until now, they appear consistent with longer trajectories, but they still need a scientific

validation that can be provided by their colocation with independent data.

**Author contribution**

A.D. developed and maintained the PET code, providing new routines and methods, generated the META3.1exp

datasets and the intermediate atlases. A.D. and C.P. developed and analyzed the similarity coefficient, provided

the result's interpretation and wrote the online documentation. C.P. wrote the manuscript, with inputs from A.D.

R.M. aided in interpreting the results and writing the manuscript. G.D. and Y.F. supervised the project.

**Competing interests**

The authors declare that they have no conflict of interest.

**Acknowledgements**

This work was funded by a CNES R&D contract.

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
