# Peer review of "META3.1exp : A new Global Mesoscale Eddy Trajectories Atlas derived from altimetry"

_Earth System Science Data, 2021_

## Author Response (AR1)

We thank the reviewers for the time spent on our paper and the given remarks.

In this comment, the reviewers' comments are in blue, our answers in black, and the modifications made in the manuscript are in orange.

All the line numbers refer to the revised manuscript.

**Answer to the Reviewer 1 comment : Citation: https://doi.org/10.5194/essd-2021-300-RC1**

The article presents a new Mesoscale Eddy Trajectory Atlas (META3.1exp) that constitutes a significant improvement over the previously-available atlas (META 2.0), which was already a landmark data set for oceanography. I found the presentation clear, and the data easily accessible and well-documented. Besides a few typos (listed below), I can only offer two suggestions about the presentation:

1. The paper makes no mention of Lagrangian Coherent Structures as an alternative set of methods for identifying coherent eddies. As a result, I felt that the literature review in the introduction was incomplete.

We agree with the reviewer, we should have mentioned those methods. We add in the introduction :

L57-61 : Moreover, a Lagrangian point of view has been used to detect Coherent Lagrangian Vortices. Different methodologies were developed to detect mesoscale eddies through direct considerations on advected fields or particles evolving together in time, in particular to reduce the number of thresholds commonly used in Eulerian methods (Abernathey and Haller, 2018; Beron-Vera et al., 2008; El Aouni, 2021; Haller et al., 2016).

**In the bibliography :**

Abernathey, R. and Haller, G.: Transport by Lagrangian Vortices in the Eastern Pacific, J. Phys. Oceanogr., 48, 667–685, https://doi.org/10.1175/JPO-D-17-0102.1, 2018.

Beron-Vera, F. J., Olascoaga, M. J., and Goni, G. J.: Oceanic mesoscale eddies as revealed by Lagrangian coherent structures, Geophys. Res. Lett., 35, L12603, https://doi.org/10.1029/2008GL033957, 2008.

El Aouni, A.: A hybrid identification and tracking of Lagrangian mesoscale eddies, Phys. Fluids, 33, 036604, https://doi.org/10.1063/5.0038761, 2021.

Haller, G.: Dynamic rotation and stretch tensors from a dynamic polar decomposition, J. Mech. Phys. Solids, 86, 70–93, https://doi.org/10.1016/j.jmps.2015.10.002, 2016.

2. While I believe I understood how "virtual observations" are used in the old tracking algorithm, it was not clear to me how they are used in the new one. (Page 7.)

The virtual observations in META3.1exp and META2.0 are similar : if during the tracking, two eddy observations are linked in a trajectory but with some time steps between them, virtual observations are used to fill the gap in the eddy detection. We slightly modify the text to remind that :

L259-264 : The tracking procedure allows us to search for a candidate eddy for up to 5 days instead of 4 (thus, a maximum of 4 consecutive virtual observations), since the overlap ensures a geographic proximity. This proximity was the only criterion in the cost function used in META2.0 in case of several candidates, whose number increases significantly when the radius of the search area and the time increase. As for META2.0, flagged virtual observations are used to fill the days without eddies within a trajectory (at most 4 consecutive virtual observations to fill the gap in detection).

**Typos:**

- 1. Line 381: "We recall that in addition of the" should be "We recall that in addition to the"
- 2. Line 387: "Thus, we produced intermediated atlases" should be "Thus, we produced intermediate atlases"
- 3. Line 491: "eddy-shaped structured" should be "eddy-shaped structures"
- 4. Line 722: "version used to product them" should be "version used to produce them"

We thank the reviewer for the typos, they have been corrected as proposed.

**Answer to the reviewer 2 comments : Citation: https://doi.org/10.5194/essd-2021-300-RC2**

In this manuscript, the authors present a new global atlas of ocean mesoscale eddies detected and tracked from satellite altimetry. This new product is a major evolution of the META2.0 atlas developed by CLS with support from CNES and distributed by AVISO+.

The changes in the algorithm of detection and tracking can be summarized as follows:

- The detection algorithm used is, for META3.1exp the Py-Eddy-Tracker (PET) developed by Mason et al., 2014) whereas META2.0 is based on an evolution of Chelton et al., (2011) algorithm.
- The detection algorithm is applied, in META3.1exp, to the Absolute Dynamic Topography (ADT) fields instead of on the Sea Level Anomaly (SLA) maps for META2.0.
- A pre-processing large scale filter is applied for both versions of the algorithm on the original altimetry fields: for META3.1exp the filter is a 2D Lanczos with cutoff wavelength set at 700 km instead of 1000 for META2.0.
- A minima of 0.4 cm of amplitude is accepted for eddies detected in META3.1exp whereas is 1 cm for META2.0.
- A shape errors is performed on the identified eddies with shapes too different from circles in META3.1exp.
- Concerning the tracking of eddies, META3.1exp bases it on overlapping between eddy areas sampled at consecutive times whereas META2.0 just look for the closest and more similar eddy in a given area.

I find the new method and eddy dataset very well described and of high potential interest for the scientific community of oceanographers and modelers. However, I would like to suggest to improve the discussion on the various setting of parameters as well as enrich the impact of the product that will be distributed by AVISO+ with a bit more validation of the dataset going beyond the mere comparison with the previous Atlas. Indeed, the manuscript is well written and the comparison between the two atlases and almost every new choice made in implementing META3.1exp from META2.0 are well documented (including the application of the algorithm to either the all-merged versus 2-satellite altimetry maps). Yet, it is very difficult to assess from the manuscript itself, the realism of one or another algorithm or the real impact of the parameter setting beyond a statistical measure of the number of eddies, their amplitude, radius and the associated trajectories.

We thank the reviewer for the remarks and hope our answers will be satisfying. As producers of the datasets, we aim to make improvements that suppress the major flaws evidenced by the scientific community, such as the detection now made with ADT instead of SLA fields, the reduction of the amplitude threshold, and the suppression of the multi-cores eddies. Providing the eddies' contours is a technical improvement that was expected by the users. With the use of the product by the scientific community, we are able to identify the changes required to represent the dynamic of the mesoscale eddies as observed and described in the recent literature. Future adjustments and evolutions will be based on the feedback of the users.

**Hence I would propose the following implementations in the discussions:**

**1. A justification of the various META3.1exp parameters setting:**

We address the following comments without strong modifications of the manuscripts. As the META3.1exp are global experimental datasets, we didn't make extensive sensibility tests on each parameter and threshold we use, that often interact with each other, and look into details as we did for META2.0 and META3.1exp major changes. Nevertheless, we did check visually and with rapid statistics the parameters during the development of META3.1exp. In the paper we present the final results and the comparison with the previous version, not the inconclusive tests.

We believe that our thresholds are adapted to the detection of the robust mesoscale eddies. Indeed, they might need adaptation for users focusing on the smaller mesoscale, or working in regions where the altimetry maps, the low levels of eddy kinetic energy, the dynamics and the geography do not allow a blind confidence in the structures detected and tracked in the META3.1exp datasets.

1. Why there is a need of a large-scale Lanczos filter on ADT? I understand that this might filters out large blobs of ADT which are not eddies, yet, such detections, if they exist, can be filtered out at a later stage either by looking at trajectories or by limiting the diameter of the eddies to a given value that need to be tested. In particular, what are the results of META3.1exp when this filter is not applied?

The filtering step aims to remove the large scale-gradients contained both in the SLA and the MDT and the equatorial "blobs" structures. With SLA input maps, Chelton et al. (2011, see Fig. 1) showed that the filter removed both the chevron-like patterns associated with the  $\beta$  refracted Rossby-waves and the steric effect due to heating, cooling, El Niño events etc (see L176-179 in the manuscript, and the 3.1.2 section). We also need to filter out the large-scale gradients in the MDT and deal with the sharp gradients of MDT in the strong currents regions such as the Gulf Stream or the Kuroshio. When filtering the MDT, we expect to filter out large-scale structures with eddy-shapes (ellipsoidal high and lows for example) that are larger than the mesoscale, and keep structures with mesoscale dimensions in the filtered map. Filtering with a 1000 km half-power cutoff wavelength maintained too large-scale structures in the filtered map, and 700 km was better to separate the large-scale in the MDT, that is why we conclude that "With this filtering, we reduced the number of large-scale eddy-shaped structures that can be detected in the low frequency MDT and maintained the dimensions of the remaining eddy-shaped structures are presented in Figure 7. The large-scale patterns in the equatorial band are efficiently removed and the strong currents gradients are preserved." (L476-476).

With no filtering, the large scale gradients are maintained, and it is more difficult to close contours satisfying the other detection's criteria (particularly amplitude) in gradients. Removing the large scale-gradient rebases the local extrema on a neutral level. In the Antarctic Circumpolar Current, where the large-scale meridional gradient is particularly sharp, more eddies are detected with the filter than without. We made an example for this situation in the py-eddy-tracker online documentation (https://py-eddy-

tracker.readthedocs.io/en/latest/python module/02 eddy identification/pet eddy detection ACC.html). Eddies can be missed or detected with smaller dimensions in non-filtered maps than without the large scale gradient (see Fig. 1 below). We saw a reduction of eddy's detection and characteristics on tests we made on a short set of detections in the South Atlantic. Moreover, as the code inherits from Chelton et al (2011) and Mason et al (2014) that both applied a large scale filtering on the input maps, our goal was to adjust for the best the cutoff wavelength. For these reasons, we did not try to build an atlas without filtering the input maps. Nevertheless, the py-eddy-tracker algorithm can be tuned by the users interested in eddy detection without filtering the large scale.

Figure 1 : Filter applied on the Absolute Dynamic Topography for an all-satellites map. a) initial ADT map, b) resulting high-pass filtered ADT used for the META3.1exp detection scheme.

**Modification in the revised manuscript :**

L457-458: The filter also rebases the local extrema on a neutral level, which helps to close contours present in these large scale patterns.

L699-703 : Note that each threshold and parameter (input maps, cutoff wavelength, amplitude, shape error, number of pixels, trajectory lifetime) used to produce the META3.1exp datasets can be adapted by the users if

they found the datasets not suitable for their study. The flexibility of the detection and tracking schemes is a strong advantage of the py-eddy-tracker package.

**2. Why an amplitude of 0.4 cm has been chosen? What is the impact of a higher or lower threshold?**

Faghmous et al. (2013) demonstrated that the 1 cm amplitude threshold (and the associated 1 cm step between SSH levels) used in Chelton et al. (2011) and followed in other studies lead to the underestimation of the eddies' properties as the step is large and the geographical extent of the structures will change a lot between two steps. Tian et al (2019) choose a smaller amplitude threshold of 0.25 cm but with contours spaced by this value, meaning the smallest eddies have two SSH contours. Small eddies were thus detected. Here we use a step between contours of 0.2 cm, and we reduced the amplitude threshold from 1 cm to 0.4 cm. This threshold implies a minimum of three SSH closed contours around an extremum to consider the area as an eddy, which seems a reasonable compromise between the time of computation and the size of the detected eddies compared to the noise and the resolution of the altimetry maps. These practical reasons led to the 0.4 cm threshold.

Nevertheless, we want to refine the amplitude threshold for the future. Thus, we recently made a comparison with a 5 years dataset where the step between contours was reduced to 0.05 cm and the amplitude threshold was thus 0.1 cm, with the META3.1exp allsat detection and tracking scheme. Here are some preliminary results :

- As the size of the eddies is restricted by the 5 pixels at least encompassed by a contour, the regions where new eddies with the 0.1cm threshold were detected are the equatorial band (± 15° latitude) for the majority of the new eddies, the regions with low levels of eddy kinetic energy (Eastern Boundary Upwelling Systems and South of the Aleoutians for example), the very high latitudes where sea ice is present, and in the complex areas (semi-enclosed seas and Indonesians straits for example).
- The repartition between untracked eddies (1%), eddies involved in short trajectories (8%) and eddies involved in long trajectories (91%) is identical between the two datasets
- The detected eddies with amplitudes <0.4 cm connect 60% of the untracked eddies in META3.1 in short trajectories (for 30%) and long trajectories (also 30%). As the META3.1 untracked eddies are only 1% of the total detected eddies, they do not strongly modify the landscape of the previously built trajectories.</li>

The smaller eddies are detected in majority in regions where the altimetry is not fully performing (high latitudes with sea ice, small oceanic regions surrounded by land) or in the equatorial band, more investigations are needed to consider these structures reliable. We do not present these preliminary results in the paper because they are not mature enough, and concern structures that by their size, duration and location are difficult to validate. To the contrary, it appears necessary to highlight the other changes that affect the representation of larger and longer eddies known to be wrongly represented in the META2.0 dataset.

We encourage the users to change the detection thresholds in the py-eddy-tracker python module to enhance the eddy detection if their studies are directly impacted by the amplitude threshold.

**Modification in the revised manuscript :**

**L187: with a 2 mm step**

L199-204: Moreover, Faghmous et al. (2013) demonstrated that the 1 cm amplitude threshold and the associated 1 cm step between SSH levels used in Chelton et al., (2011) and followed in other studies lead to the underestimation of the eddies' properties as the step is large and the geographical extent of the structures will change a lot between two steps. The 0.4 cm threshold implies a minimum of three SSH closed contours around an extremum to consider the area as an eddy, which is a reasonable compromise between the time of computation and the size of the detected eddies compared to the noise and the resolution of the altimetry maps.

3. What is the impact of the shape test error? Does it really impact the final results?